REGISTERED REPORT PROTOCOL

# Registered report: The effectiveness of a Bhagavad Gita intervention to reduce psychological distress in homeless people—A randomised controlled trial

**Laalithya Konduru**[1,2,3]\*, **Simranjeet Singh Dahia**[1,4], **Gargi Kothari-Speakman**[1]

**1** Savitri Ghantasala Center for Health Equity, Samanjasa Foundation, Chennai, Tamil Nadu, India, **2** Department of Community Medicine, Sri Jagannath Healthcare and Research Center, Dhanbad, Jharkhand, India, **3** College of Medicine and Public Health, Flinders University, Adelaide, South Australia, Australia, **4** School of Computer Science, The University of Adelaide, Adelaide, South Australia, Australia

\* laalithya@gmail.com

## Abstract

### Introduction

The coronavirus disease pandemic has worsened psychological distress in people experiencing homelessness (PEH). This study evaluates the impact of learning the Bhagavad Gita versus engaging in Kuchipudi dance on reducing psychological distress in PEH in Chennai and Dhanbad, India.

### Methods and analysis

Participants will be allocated into Bhagavad Gita, Kuchipudi dance, Both interventions, or Control groups. The Kessler scale (K10) score, representing the primary outcome, will be measured at four time points: one day before intervention, one day after intervention completion, 40 days post-intervention, and 1 year post-intervention. The K10 scores will be analyzed using the Generalized Estimating Equation framework. Additionally, subgroup analysis based on participant demographics (e.g., age, education, religion, employment) will be conducted to explore potential differential effects using analysis of covariance. Statistical significance will be determined at $p < 0.05$ (two-tailed).

### Dissemination

After study completion, findings will be shared at conferences, in peer-reviewed journals, and with stakeholders and community groups. Authorship will be granted to contributing researchers, with acknowledgment for others. The dataset will be publicly available upon publication. Efforts will be made to communicate results through infographics, plain language summaries, and tailored strategies, including community meetings and digital platforms, to engage and empower PEH in decision-making.

**Data Availability Statement:** All relevant data from this study will be made available upon study completion.

**Funding:** The authors received no specific funding for this work.

**Competing interests:** The authors have declared that no competing interests exist.

## Trial registration

The trial has been registered with the Clinical Trials Registry of India (registration number: CTRI/2022/12/048416).

## Introduction

The COVID-19 pandemic has had a profound impact on the mental health of people experiencing homelessness (PEH) [1, 2] PEH have reported increased rates of anxiety, depression, and post-traumatic stress disorder [1, 2], with a significant proportion suffering from severe mental illness and chronic substance use [2]. Moreover, social isolation during the pandemic has been associated with heightened anxiety and depression symptoms among PEH [2]. Substance use, which is prevalent in this population, can further worsen mental health outcomes [2, 3].

The World Health Organization (WHO) defines mental health as a state of well-being, encompassing an individual's ability to realize their potential, cope with life's challenges, work productively, and contribute to their community [4]. Recognizing the importance of mental health, the WHO has advocated for integrating mental health support into the global response to the COVID-19 pandemic [5]. General guidance from the WHO on integrating mental health support into the pandemic response which can be applied to PEH includes providing social support, accurate information, and addressing stigma and discrimination.

Samanjasa Foundation has been actively engaged in providing services to PEH in Chennai, Dhanbad, Bhubaneshwar, and Hyderabad, India, for the past decade. These services include offering meals, upskilling opportunities, recreational activities, and linkages to government schemes and employment prospects. During the COVID-19 lockdown in India, we continued to support PEH by providing online recreational and upskilling activities. However, due to logistical challenges, we were unable to resume face-to-face sessions after the lockdown was lifted. Throughout our interactions with PEH, we observed an increase in psychological distress during and after the pandemic. To address this issue, we decided to augment our usual recreational activities, such as Kuchipudi dance classes [6], with an emic intervention involving imparting the teachings of the Bhagavad Gita through online platforms [7].

Engaging in hobbies has been shown to effectively reduce stress, foster social connections, and restore a sense of self among PEH [8, 9]. However, many of our clients have been unable to participate in recreational activities due to their daily struggles to earn a living, despite expressing a desire to engage in hobbies. Therefore, we sought a brief intervention that could enhance the coping abilities of PEH.

Findings from a previous study revealed that turning to religion emerged as a significant coping strategy for PEH, assisting them in managing the mental stress induced by the COVID-19 pandemic [10]. Building on this crucial insight, we posit that a religious and spiritual Intervention (RSI) could offer valuable support to PEH.

A notable systematic review and meta-analysis focused on RSIs in randomized clinical trials (RCTs) has underscored the potential of RSIs in enhancing mental health outcomes; however, this review exclusively encompassed studies affiliated with Islamic, Christian, and Jewish traditions [11]. The meta-analysis unveiled significant positive effects of RSIs on anxiety, general symptoms, and depressive symptoms. Nonetheless, due to the omission of any clinical trials focusing on the efficacy of Hindu RSIs or the impact of RSIs on individuals adhering to Hindu

beliefs, a gap persists in understanding the benefits of Hindu RSIs, particularly among those embracing Hindu beliefs.

It is noteworthy that Hinduism is the most commonly practiced religion in India and there exist no empirical studies evaluating the effectiveness of Hindu RSIs. Therefore, tailoring an RSI to the Hindu context, especially in PEH, in the absence of empirical studies investigating the impact of Hindu RSIs, is a critical knowledge gap. Given the significant reliance on religious coping mechanisms among PEH [10]. and the dearth of research in this domain, our study aims to fill this void by specifically exploring the impact of learning the Bhagavad Gita—an essential Hindu scripture—as a potential RSI to alleviate psychological distress among PEH. This novel approach is informed not only by previous qualitative findings [10] but also by the imperative to address the unique needs of this population through culturally relevant interventions.

The Bhagavad Gita offers practical guidance on navigating daily stressors [12]. Within the realm of RSIs, recent studies offer insights into the potential relevance of the Bhagavad Gita's teachings for psychological well-being among adolescents [13] and individuals coping with chronic illnesses [14]; however, these studies do not encompass a synthesis of empirical data to validate their findings. Recently, Das and Behura proposed that learning the Bhagavad Gita could improve the stress coping abilities of healthcare providers during the COVID-19 pandemic, drawing parallels between their experiences and those of war veterans; however, this study also lacked empirical substantiation [15].

Considering that homelessness also poses significant challenges to human health, including mental health, and our organization's volunteers had noted that a significant number of PEH had turned to spirituality as a coping mechanism to deal with the mental health challenges brought about by the COVID-19 pandemic, we hypothesized that imparting the teachings of the Bhagavad Gita as a one-time intervention could enhance the psychological resilience of PEH.

This study aims to bridge the knowledge gap concerning the effectiveness of Hindu RSIs in a previously unexplored population—PEH—by evaluating the impact of learning the Bhagavad Gita on reducing psychological distress among PEH. Engaging in Kuchipudi dance practice as a hobby will function as a positive control, while abstaining from stress-reducing activities will serve as the negative control. In other words, the central research question is, what is the impact of learning the Bhagavad Gita on reducing psychological distress among PEH, and how does this compare with the impact of engaging in Kuchipudi dance practice as a hobby? This multi-site parallel-group RCT seeks to comprehensively assess both short and long-term outcomes. Table 1 outlines the research inquiries guiding our study and presents potential interpretations of the anticipated results.

## Materials & methods

### Ethical considerations

The study adhered to the SPIRIT reporting guidelines (S1 Checklist). Approval for the study was granted by the Sri Jagannath Healthcare and Research Center–Independent Ethics Committee (approval number: SJHRC–N4/22/OCT/07), and it was registered with the Clinical Trials Registry of India (registration number: CTRI/2022/12/048416). The protocol described herein is version 1 dated December 23, 2022. Changes to the protocol arising from peer review will be updated as version 2 and appropriate updates will be made to the protocol on the Clinical Trials Registry of India; the ethics committee will also be apprised of the changes and provided a copy of the revised protocol. Participants will provide written informed consent, and their next of kin will provide oral informed consent before commencement of their participation.

## Participants

Persons meeting the criteria for the definition of a houseless person—someone who does not live in a building or "census houses," but rather in the open or seeks shelter in public places such as railway stations, under flyovers, places of worship, etc.—according to the 2011 Census of India, and residing in Chennai or Dhanbad, India during and after the Unlock 2.0 phase of the COVID-19 pandemic in India, aged >18 years will be recruited. Persons with a formal diagnosis of anxiety, depression, and any other mental illness will be excluded from the study

**Table 1. Study design.**

| Question | Hypothesis | Sampling plan | Analysis plan | Interpretation given to different outcomes |
|---|---|---|---|---|
| Does learning the Bhagavad Gita reduce psychological distress in people experiencing homelessness | The immediate post-intervention, delayed post-intervention, and long-term follow-up Kessler psychological distress scale scores will be significantly lower than baseline Kessler psychological distress scale scores | 54 participants each in the Bhagavad Gita group, Kuchipudi dance group, Both Bhagavad Gita and Kuchipudi dance group, and Control group for 80% power and 5% alpha | The within-group differences will be analysed by repeated measures analysis of covariance, with continued practice and religion as covariates. Subgroup analyses will be conducted by stratifying the participants in each group by age, religion (Hindu, Non-Hindu), educational attainment, and employment. Post-hoc power analyses will be conducted to verify the robustness and reliability of the subgroup analyses | If we find significant within-group differences in the Kessler psychological distress scale scores, it would suggest that learning the Bhagavad Gita has a potential impact on reducing psychological distress in people experiencing homelessness. Additionally, if our subgroup analyses reveal significant differences based on age, religion, educational attainment, and employment, it would imply that the intervention's effectiveness varies across these subgroups |
| Is learning the Bhagavad Gita better at reducing psychological distress than engaging in a hobby in people experiencing homelessness | The decrease from baselines to immediately post-intervention, delayed post-intervention, and long-term follow-up Kessler psychological distress scale scores will be significantly larger in the Bhagavad Gita group than in the Kuchipudi dance group | 54 participants each in the Bhagavad Gita group, Kuchipudi dance group, Both Bhagavad Gita and Kuchipudi dance group, and Control group for 80% power and 5% alpha | Intergroup differences will be analysed by the generalised estimating equation and analysis of covariance with continued practice and religion as covariates. Subgroup analyses will be conducted by stratifying the participants in each group by age, religion (Hindu, Non-Hindu), educational attainment, and employment. Post-hoc power analyses will be conducted to verify the robustness and reliability of the subgroup analyses | If our hypothesis holds true, it would suggest that learning the Bhagavad Gita is more effective at reducing psychological distress than engaging in a hobby for people experiencing homelessness. By conducting subgroup analyses, we will further investigate if the effectiveness of the intervention varies based on age, religion, educational attainment, and employment |
| Does continued practice influence the effect of the interventions in people experiencing homelessness | The decrease from baselines to delayed post-intervention and long-term follow-up Kessler psychological distress scale scores will be significantly larger in all intervention groups in participants who continue the practices learned during the intervention compared to that in participants who do not continue the practices learned during the intervention | 54 participants each in the Bhagavad Gita group, Kuchipudi dance group, Both Bhagavad Gita and Kuchipudi dance group, and Control group for 80% power and 5% alpha | Participants in each group will be subdivided based on whether or not they continued practice during each follow-up. Repeated measures analysis of variance will be conducted on the subgroups | If our hypothesis holds true, it would suggest that sustained practice contributes to greater psychological well-being. Conversely, if there are no significant differences in the decrease of the Kessler psychological distress scale scores between participants who continue and those who do not continue the practices, it would indicate that continued practice does not play a significant role in the effectiveness of the interventions |

*(Continued)*

**Table 1.** (Continued)

| Question | Hypothesis | Sampling plan | Analysis plan | Interpretation given to different outcomes |
|---|---|---|---|---|
| Does learning the Bhagavad Gita provide any added benefit to people experiencing homelessness who are engaging in a recreational activity in terms of reducing their psychological distress | The decrease from baselines to immediately post-intervention, delayed post-intervention, and long-term follow-up Kessler psychological distress scale scores will be significantly larger in the Both group than the Bhagavad Gita and Kuchipudi dance groups | 54 participants each in the Bhagavad Gita group, Kuchipudi dance group, Both Bhagavad Gita and Kuchipudi dance group, and Control group for 80% power and 5% alpha | Intergroup differences will be analysed by the generalised estimating equation and analysis of covariance with continued practice and religion as covariates. Subgroup analyses will be conducted by stratifying the participants in each group by age, religion (Hindu, Non-Hindu), educational attainment, and employment. Post-hoc power analyses will be conducted to verify the robustness and reliability of the subgroup analyses | If the reduction in the Kessler psychological distress scale scores is highest in the Both group, it would suggest that learning the Bhagavad Gita combined with engaging in a recreational activity may offer additional benefits in reducing psychological distress among homeless persons compared to learning the Bhagavad Gita group or engaging in a recreational activity alone. This finding would indicate that the combination of the intervention has a more substantial effect on decreasing distress levels |

and provided access to evidence-based treatments. All PEH availing the services of Samanjasa Foundation in Chennai and Dhanbad will be approached to participate in the study; the study information will be disseminated through the case workers. During earlier studies we found that this strategy ensures nearly 90% response rate; therefore, we are expecting a similar enrolment rate in this study as well. The participants will be divided into four groups—the Bhagavad Gita (Gita), Kuchipudi dance (Kuchipudi), both Bhagavad Gita and Kuchipudi dance (Both), and control groups—in a 1:1:1:1 ratio. The participant identification number, group assignment, four copies of a questionnaire with pre-filled participant ID and follow-up date, meeting schedule, and Zoom meeting link will be sealed in a small envelope and the envelopes will be placed in a fishbowl and shuffled. The participant ID will be generated in a random order by an independent statistician using a pseudo-random number generator script in Python 3.9.13. Group assignment will be generated in sequence by the independent statistician; the participant ID generated first will be associated with the Gita group, the second with the Kuchipudi group, the third with the Both group, the fourth with the Control group, the fifth with the Gita group, and so on. The project staff will fill and seal the envelopes with the respective contents. Once a participant provides written informed consent to participate in the study, they will be asked to pull an envelope from the fishbowl and not reveal its contents to the research team. They will also be instructed not to read the questionnaires until the sessions scheduled for the purpose of filling them out. To monitor and minimize the risk of contamination, especially given the potential for participants within the same community to communicate, the project staff will send weekly text messages to the participants regarding not sharing details about the intervention with anyone, including their family members. Additionally, the project staff will randomly contact three participants from each group via a phone call every two months and ask them if they had heard anything about the interventions in the other study groups. Furthermore, during community events and interactions, case workers will be asked to assess the level of knowledge regarding the intervention among the PEH. The participant journey is illustrated in Fig 1.

## Interventions

The participants in the Gita group will be imparted the teachings of the second chapter of the Bhagavad Gita (all verses) for 1 week, for 60 minutes per day. The teachings of the Bhagavad

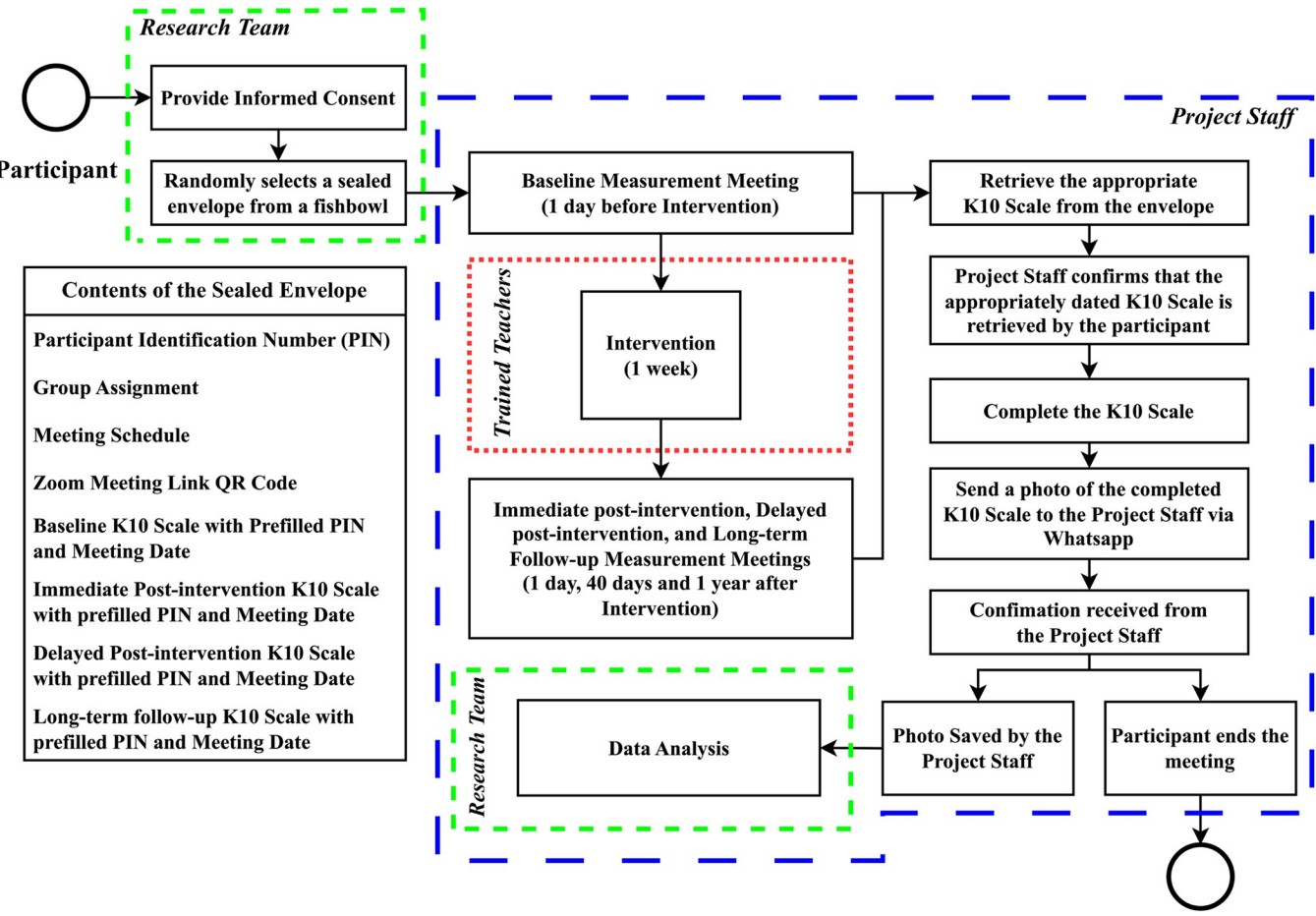

**Fig 1. Participant journey: Navigating through the study.**

Gita will be imparted by a trained teacher and the second chapter has been chosen as it is said to contain the essence of the entire Bhagavad Gita [16]. The participants in the Kuchipudi group will be taught the Kuchipudi dance choreography to a popular devotional song ("Sivashtakam", choreographed by Padma Bhushan Dr. Vempati Chinna Satyam, song duration is approximately 10 min) for 60 minutes per day by a trained teacher for 1 week. Every day for 1 week, the participants in the Both group will be taught the Kuchipudi dance choreography to a popular devotional song ("Vandanamu Raghunandana", choreographed by Kala Tilakam Dr. Sathyapriya Ramana, song duration is approximately 5 min), which is shorter in duration than the song used in the Kuchipudi group, for 30 minutes by a trained teacher, followed by imparting the teachings from selected verses of the second chapter of the Bhagavad Gita (Verse 40–72) for 30 minutes by a trained teacher. The participants in the control group will receive no intervention. The participants will be instructed to join their assigned groups at the scheduled meeting times using the meeting link in their sealed envelopes. Apart from the teaching sessions, meetings will also be scheduled one day before the start of the teaching sessions, one day after the end of the teaching sessions, 40 days after the end of the teaching sessions, and 1 year after the end of the teaching sessions for the purpose of recording the baseline, immediate post-intervention, delayed post-intervention, and long-term follow-up measurements, respectively. For participants in the control group, the meeting schedule will consist of only the four measurement sessions.

## Outcomes and measures

The primary outcome is psychological distress measured by the Kessler scale of psychological distress (K10) in this study. The K10 is a 10-item questionnaire that measures psychological distress based on questions about anxiety and depressive symptoms. It has previously been used in research conducted with PEH [17, 18]. At the immediate post-intervention, delayed post-intervention, and long-term follow-up measurements, apart from the K10, the participants will be asked if they have continued practicing the techniques learned during the intervention (Kuchipudi group: continued practice of the choreography, Gita group: continued self-reflection on the teachings of the Bhagavad Gita, Both group: continued practice of the choreography and continued self-reflection on the teachings of the Bhagavad Gita, and control group: none). The study flowchart is presented in Fig 2.

## Data collection and management procedures

During each scheduled measurement session, the participants will be asked to take the questionnaire they were provided with in the sealed envelopes given to them at the time of enrolment that is pre-filled with the date of the appropriate measurement session. They will then be asked to hold the questionnaire sheet with the date visible to the camera and trained research support staff will check that each participant is holding the correct questionnaire. They will then walk the participants through how to fill in the questionnaire. Once they confirm that the questionnaires have been filled, they will walk the participants through how to take a photograph of the sheet and send the photo to the staff member via WhatsApp. Once the staff member receives legible photographs of the filled questionnaires from all participants, the meetings will be ended. The project staff will then save the photographs into as portable document format files on a secure computer situated in the research office of our organization and print out the same for our records. Our organization's policy is to store both soft and hard copies of all research material—the soft copies are stored for a period of 2 years and the hard copies are stored for a period of 10 years. The research team will have unrestricted access to the final dataset.

## Statistical methods

**Data analysis.** All results will be reported in accordance with the CONSORT guidelines to ensure comprehensive and transparent reporting of the trial's outcomes. Descriptive summaries of baseline characteristics will be presented to provide an overview of the study population. The comparisons will be between each intervention group and the control group, as well as within each intervention group over time. The repeated measures data will be analysed using the generalised estimating equation (GEE) framework. For the analysis of the primary outcome using the GEE framework, robust standard errors will be used to account for correlated data. Pairwise post-hoc comparisons will be conducted to identify specific group differences. To control for the risk of type I error in these multiple comparisons, the Bonferroni correction will be applied. Subgroup analysis will be conducted using analysis of covariance (ANCOVA) or non-parametric ANCOVA [19] depending on the normality of the data; this may include stratification by age, educational attainment, religion, and employment status. Multiple comparisons will also be addressed using appropriate statistical corrections. These may include the Bonferroni correction, Holm's adjustment, or the Benjamini-Hochberg procedure, depending on the specific comparisons and the number of tests conducted. The data analysis will be conducted using Python 3.9.13. Statistical significance will be set at a two-tailed $p < 0.05$.

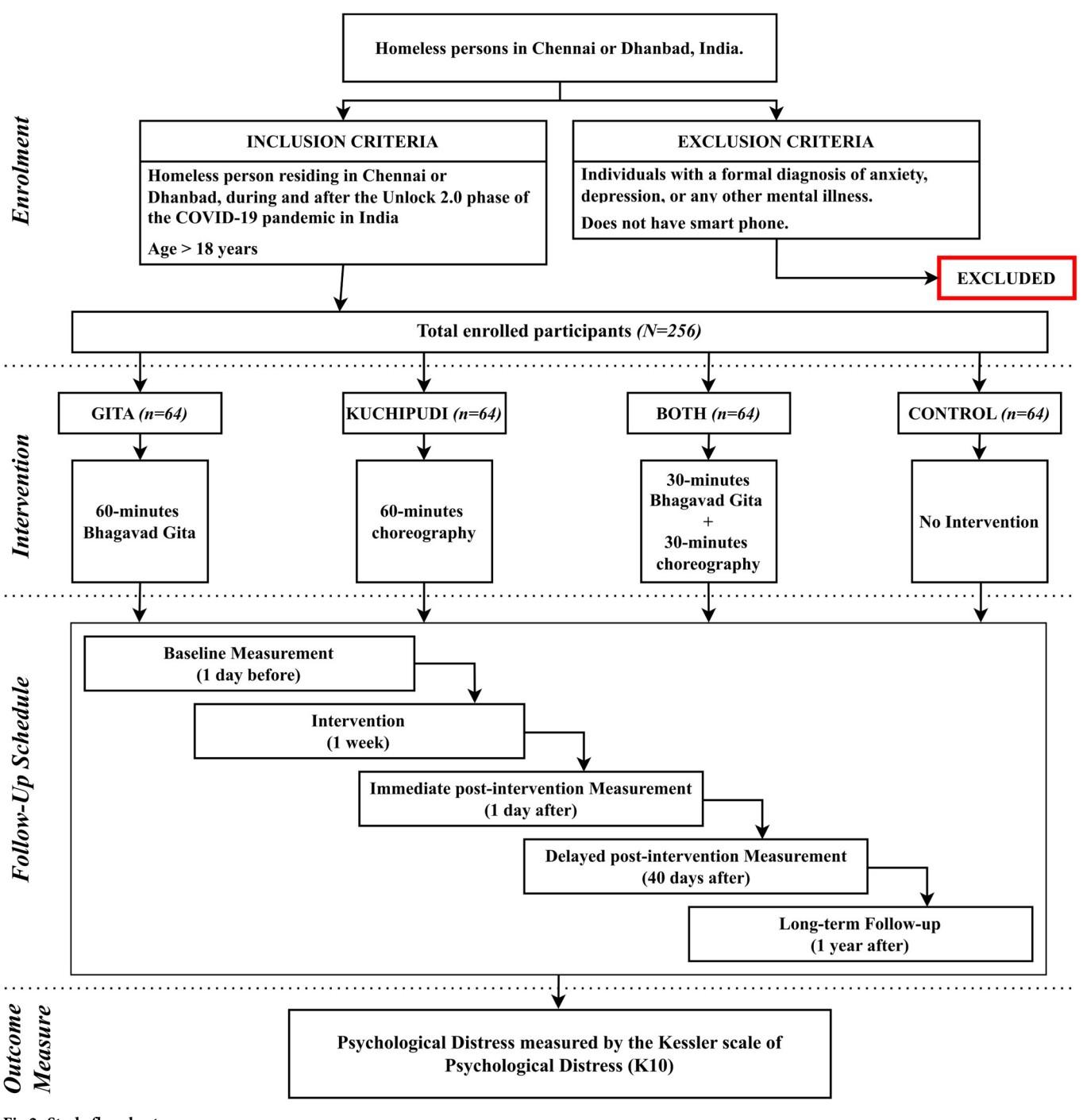

**Fig 2. Study flowchart.**

**Sample size calculation.** The required sample size for GEE was calculated using the formula,

$$n = [(Z\alpha/2 + Z\beta)^2 * V\_eff]/(\Delta^2 * \rho^2),$$

where n is the required sample size per group; $Z\alpha/2$ is the critical value of the standard normal

distribution for a two-tailed test with a significance level of α; Zβ is the critical value of the standard normal distribution for a specified power (1-β) of the test; V_eff is the effective sample size, which accounts for the correlation between the repeated measures within individuals; Δ is the minimum clinically significant difference; r is the number of repeated measures; and ρ is the correlation between the repeated measures.

Assuming a conservative value of 0.5 for ρ based on previous literature [20, 21], and for an α of 5%, 1-β of 80%, r of 4, and Δ of 7 [22, 23], we can calculate the effective sample size as follows:

$$V\_eff = n/(1 + (r-1)*\rho)$$
$$= 55/(1 + (4-1)*0.5)$$
$$= 22.5$$

Therefore, n can be calculated as follows:

$$n = [(1.96 + 0.84)^2 * 22.5]/(7^2 * 0.5^2)$$
$$= 53.3$$

Thus, at least 54 participants per group must be enrolled to have 80% power to detect a minimum clinically significant difference of 7 points between the groups using GEE, with a significance level of 5%. As there are four groups in the study, a total of 216 participants (54 per group) must be enrolled. Based on our previous experience, a dropout rate of 16% was expected; to account for this, we aim to recruit a total of 256 participants, with 64 participants per group.

## Blinding

The study will involve three separate teams—trained teachers, project staff, and the research team. Trained teachers will be responsible for administering the interventions; the teachers will be blinded to the K10 results. Project staff will be responsible for ensuring participant compliance, collecting the filled K10 questionnaires as photos via WhatsApp, and printing the photos and storing them until data analysis; project staff will be blinded to data analysis. The research team will be responsible for study design, participant recruitment, obtaining written informed consent, data analysis and reporting; the research team will be blinded to participant group allocation.

### Plan for handling missing data and protocol deviations

In case we are unable to recruit the required number of participants for the study from only Chennai and Dhanbad during the envisaged recruitment period, we will open the recruitment to the remaining locations that Samanjasa Foundation operates in, i.e., Hyderabad and Bhubaneshwar, India, and extend the recruitment period by another 4 weeks. Consequently, the timelines for the baseline measurements, interventions, and all post-intervention measurements will be delayed by 4 weeks. To minimize missing data, the participants will be given clear written instructions in their mother tongue and the project staff will regularly follow-up with the participants to ensure that they will not drop out of the study. Before the scheduled meetings, the participants will receive text messages and a phone call reminder from the project staff. Depending on post-hoc power analysis results, missing data will be handled either by complete case analysis or multiple imputation. Protocol deviations may result from technology failure and failure of the participants to comply with the interventions. The clear written instructions to the participants in their mother tongue and regular follow-up by project staff

will help facilitate participant compliance. In case of technology failure, the project staff will video call the participants through WhatsApp, as the participants are more familiar with it than with Zoom. To prevent technology failure due to running out of data, all participants will receive an unlimited data recharge with one month validity one day before the scheduled baseline, delayed post-intervention, and long-term follow-up measurements. This can also serve as an incentive to the participants to attend the scheduled measurement sessions and help minimize missing data. In case of technology failure due to other reasons, the participants will be offered catch-up sessions before the next scheduled session.

## Statistical analysis plan (SAP)

This SAP outlines the procedures for data cleaning, handling missing data, conducting primary and subgroup analyses, and performing sensitivity analyses to ensure the robustness of the findings.

### Data cleaning

1. Data Entry and Validation: All data will be entered into a secure electronic database. Double data entry will be performed to ensure accuracy. Discrepancies will be resolved by consulting the original data source.

2. Consistency Checks: Logical checks will be conducted to identify any inconsistencies (e.g., age values outside the expected range). Outliers will be flagged and verified with the original data.

3. Descriptive Summaries: Baseline characteristics of participants will be summarized using means and standard deviations for continuous variables and frequencies and percentages for categorical variables.

### Handling of missing data

1. Identification: Missing data will be identified and categorized (e.g., missing completely at random, missing at random, or not missing at random).

2. Imputation: Multiple imputation methods will be used to handle missing data, particularly if the proportion of missing data exceeds 5%. Sensitivity analyses will be conducted to compare the results from imputed and non-imputed datasets.

3. Documentation: The extent and nature of missing data will be documented and reported.

### Primary analysis

1. GEE Framework: The primary outcome, psychological distress as measured by the K10 scale, will be analyzed using the GEE framework. Robust standard errors will be used to account for correlated data due to repeated measures. An exchangeable correlation structure will be assumed.

2. Primary Comparisons: Comparisons will be made between each intervention group (Gita, Kuchipudi, and Both) and the control group. Within each intervention group, comparisons over time (pre-intervention, immediate post-intervention, 40 days post-intervention, and 1-year post-intervention) will be performed.

3. Pairwise Post-Hoc Comparisons: Pairwise post-hoc comparisons will be conducted following significant GEE results to identify specific group differences. The Bonferroni correction will be applied to control for the risk of type I error in these multiple comparisons.

## Subgroup analyses

1. ANCOVA/Non-Parametric ANCOVA: Subgroup analyses will be conducted using ANCOVA or non-parametric ANCOVA depending on the normality of the data. Subgroups may include stratification by age, educational attainment, religion, and employment status.

2. Multiple Comparisons: To address the risk of type I error due to multiple comparisons, appropriate statistical corrections will be applied, which may include the Bonferroni correction, Holm's adjustment, or the Benjamini-Hochberg procedure.

## Sensitivity analyses

1. Robustness Checks: Sensitivity analyses will be performed to assess the robustness of the primary findings. Analyses will include different imputation methods for missing data and alternative statistical models (e.g., mixed-effects models).

2. Assumption Testing: Assumptions of the GEE and ANCOVA models will be tested, including the assumption of normality and homogeneity of variances. Alternative models or transformations will be used if assumptions are violated.

3. Exclusion of Outliers: Sensitivity analyses will be conducted with and without outliers to determine their impact on the study results.

**Adjustment and interpretation based on contamination checks.** If contamination is detected, we will take the following steps to adjust or interpret the study results:

1. Quantification of Contamination: We will quantify the extent of contamination by calculating the percentage of participants who have been exposed to details about interventions other than their own.

2. Statistical Adjustment: We will use statistical techniques to adjust for contamination. For instance, we might include contamination status as a covariate in our regression models to control for its potential impact on the study outcomes.

3. Sensitivity Analyses: We will conduct sensitivity analyses to assess the robustness of our findings. This will involve comparing the results of the main analysis with those obtained after excluding contaminated participants or adjusting for contamination.

4. Interpretation of Results: In the discussion section, we will interpret our findings in light of the contamination checks. We will acknowledge the presence of contamination, describe its potential impact on the study outcomes, and discuss how our adjustments have accounted for it.

This SAP will be reviewed and updated as necessary throughout the study to ensure the robustness and integrity of the analyses and findings.

## Risk mitigation strategy

As PEH are a vulnerable group, several risk mitigation strategies have been put in place to ensure the safety and wellbeing of the participants. These are as follows:

1. Informed Consent will be obtained from each participant, after providing them with oral and written information clearly explaining the purpose, procedures, potential risks, and benefits of the intervention. To ensure that the participants understand their rights, the voluntary nature of participation, and their ability to withdraw at any time without consequences, they will be asked to repeat the information in their own words, orally and in writing, to the research team.

2. To facilitate compliance with the intervention, the project staff will be drawn from the volunteers who are involved in the day-to-day activities of our organization, engaging regularly with our PEH clients. The trained teachers who will administer the interventions will be from external organizations; this will help the participants engage with organizations other than our own, thus deepening their engagement with civil society. To maintain objectivity and minimise the possibility that the participants may feel coerced to participate in the research on account of being approached by volunteers they frequently engage with, the research team is kept separate from the client engagement teams in our organization.

3. Secure online platforms will be used in adherence to data protection regulations of our organization to safeguard personal data of all stakeholders. The smartphones used for sending and receiving correspondence and photographs of the filled K10 questionnaires will be those provided by the organization. The project staff will not use their personal phone numbers for communication with the participants under any circumstance.

4. Keeping the culture of India in mind, the participants will be asked to provide oral consent from their next of kin. This will be recorded on the consent forms and countersigned by the participant. In case the participant is unable to read or sign their name, the consent process will be conducted in the presence of two healthcare professionals who are not related to our organization. A blank informed consent form in English and Hindi has been provided as S1 File. The informed consent forms in other languages (Tamil, Telugu, and Bengali) are true translations of the informed consent form in English.

5. The participants' technological access and proficiency will be assessed at the time of obtaining consent to ensure that they can fully engage with the online intervention. In case technological proficiency needs to be improved, digital literacy training and training on the digital platforms to be used in the intervention will be provided.

## Plan for dissemination of results

Upon completion of the trial, a comprehensive plan for dissemination and publication of the results will be implemented. This includes presenting the findings at relevant conferences, submitting manuscripts to peer-reviewed journals, and sharing the outcomes with key stakeholders, policymakers, and other community organizations working with PEH. Authorship will be given to members of the research team who also contribute to writing of the manuscripts; all others will be acknowledged. The final dataset will be published in a public repository upon publication of the results of the study. Additionally, efforts will be made to communicate the findings in accessible formats, such as infographics or plain language summaries, to reach a broader audience and promote the potential impact of the intervention. Specifically tailored to PEH, community meetings and community engagement volunteers will be utilized to directly engage the PEH and share the trial results in relatable and accessible formats including videos and memes. Peer educators will be trained to disseminate information and digital platforms such as WhatsApp will be leveraged to provide informational materials, empowering PEH to make informed decisions about their wellbeing.

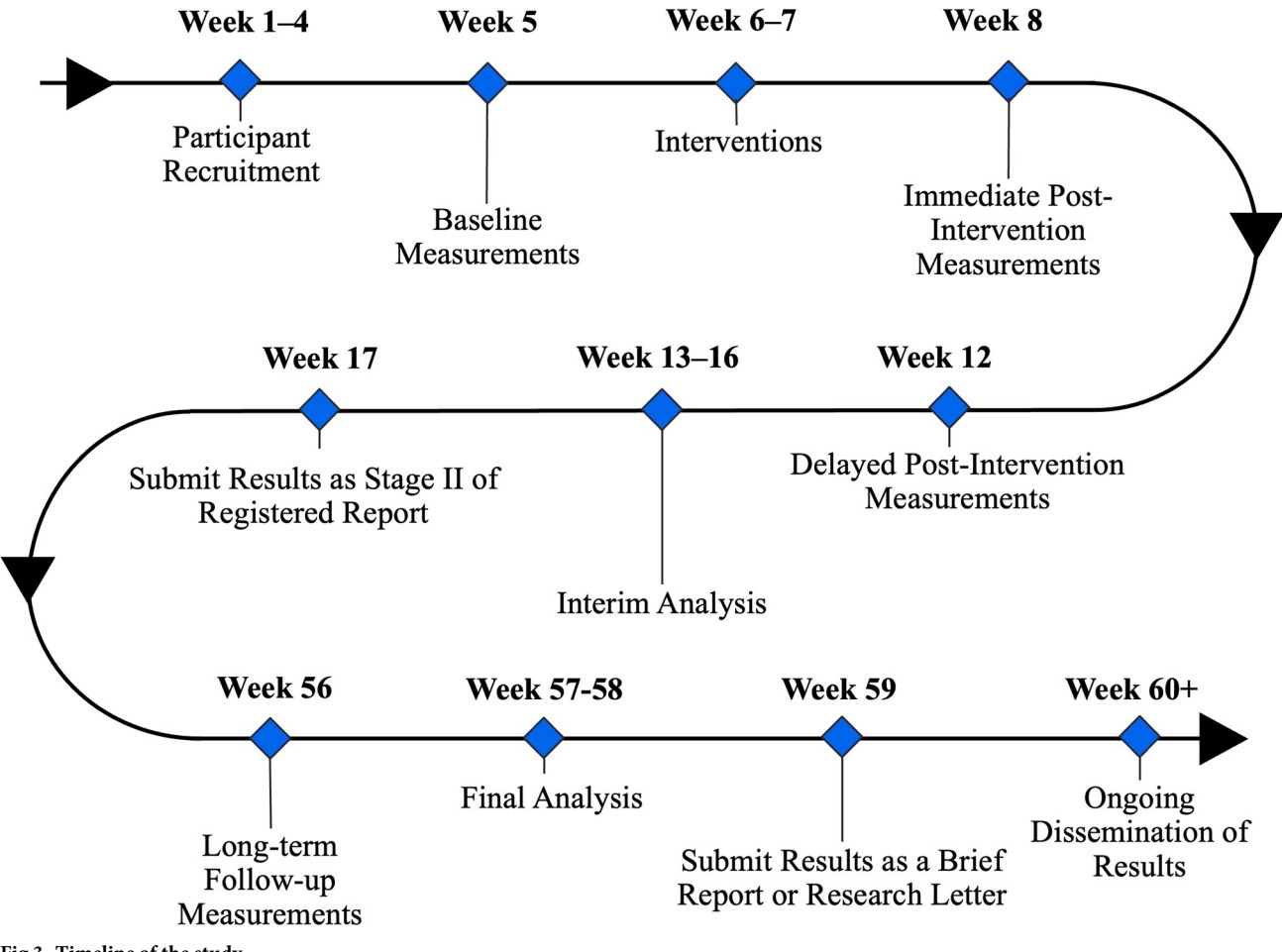

**Fig 3. Timeline of the study.**

## Timeline

Engagement with PEHs begins in Week 1, followed by a 4-week recruitment period. Baseline measurements will be conducted in Week 5, interventions in Weeks 6–7, and post-intervention assessments in Weeks 8, 12, and 56. Interim analysis shall be conducted in Weeks 13–19, and results will be published as the Stage II of this registered report. Final analysis will be conducted in Weeks 57–61, and results shall be published as a brief report. Conferences, meetings, workshops, and other community engagement events to disseminate the results of the study will be ongoing indefinitely, starting from the date of publication of the interim analysis as a registered report. The detailed timeline is shown in Fig 3.

## Patient and public involvement statement

Patients and the public were engaged throughout the research process, starting from its inception. The Samanjasa Foundation's longstanding involvement in supporting PEH in Chennai, Dhanbad, Bhubaneshwar, and Hyderabad over the past decade facilitated a deep understanding of PEH's challenges and needs. Additionally, an interpretative phenomenological analysis conducted in this population revealed that they had turned to spirituality as a coping mechanism during the COVID-19 pandemic [10]. This informed the development of research questions and outcome measures, aligning them with priorities, experiences, and preferences of

PEH. The study's design was shaped by their input, leading to the integration of both an emic intervention involving the Bhagavad Gita teachings and a positive control using Kuchipudi dance. Recruitment strategies were informed by PEH's circumstances and perspectives, ensuring their meaningful involvement in the research. Participants were consulted to assess the intervention burden and time commitment, resulting in interventions that accommodate their daily struggles. The comprehensive dissemination plan, including presentations at conferences, sharing outcomes with stakeholders, and leveraging accessible formats, aims to engage PEH and wider patient communities in the dissemination of study results. Strategies such as community meetings, digital platforms, and tailored approaches ensure results are communicated effectively, respecting their preferences and empowering them to make informed decisions about their well-being based on study findings.

## Strengths and limitations of this study

The study takes into account the complex impact of the coronavirus disease pandemic on mental health among people experiencing homelessness (PEH), addressing a timely and critical need. The incorporation of both a traditional emic intervention (Bhagavad Gita teachings) and a hobby-based positive control (Kuchipudi dance) demonstrates a multifaceted approach to enhancing coping mechanisms for PEH. Furthermore, the involvement of a Non-Governmental Organization with a history of dedicated support for PEH lends credibility and ensures the study is grounded in real-world experiences and needs. While the Kessler scale is a validated tool for assessing psychological distress, relying solely on self-report measures could introduce response bias and potential inaccuracies due to participants' subjective interpretations. Moreover, the emic intervention involving the Bhagavad Gita teachings might not be universally applicable to all PEH due to cultural differences and varying beliefs, potentially impacting the intervention's effectiveness and relevance.

## Supporting information

**S1 Checklist. SPIRIT 2013 checklist: Recommended items to address in a clinical trial protocol and related documents.**
(DOC)

**S1 File. Consent form in English and Hindi.**
(DOC)

## Acknowledgments

We would like to acknowledge Narthanasaala School of Kuchipudi Dance, Chennai for agreeing to teach the Kuchipudi dance choreography and Dr. Nishant Das of the Department of Humanities and Social Sciences, Indian Institute of Technology (ISM), Dhanbad, Jharkhand, India for agreeing to teach the relevant verses from the Bhagavad Gita.

## Author Contributions

**Conceptualization:** Laalithya Konduru.

**Data curation:** Simranjeet Singh Dahia.

**Formal analysis:** Laalithya Konduru, Simranjeet Singh Dahia, Gargi Kothari-Speakman.

**Investigation:** Laalithya Konduru, Simranjeet Singh Dahia, Gargi Kothari-Speakman.

**Methodology:** Laalithya Konduru.

**Project administration:** Laalithya Konduru.

**Resources:** Gargi Kothari-Speakman.

**Software:** Simranjeet Singh Dahia.

**Supervision:** Laalithya Konduru.

**Validation:** Laalithya Konduru, Gargi Kothari-Speakman.

**Visualization:** Simranjeet Singh Dahia.

**Writing – original draft:** Laalithya Konduru, Simranjeet Singh Dahia.

**Writing – review & editing:** Gargi Kothari-Speakman.

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
