## [Decision Letter · Decision Letter 0]

5 Dec 2023

PONE-D-23-29874Registered Report: The Effectiveness of a Bhagavad Gita Intervention to Reduce Psychological Distress in Homeless People—A Randomised Controlled TrialPLOS ONE

Dear Dr. Konduru,

Thank you for submitting your manuscript to PLOS ONE. After careful consideration, we feel that it has merit but does not fully meet PLOS ONE’s publication criteria as it currently stands. Therefore, we invite you to submit a revised version of the manuscript that addresses the points raised during the review process.

The manuscript has been evaluated by two  reviewers, and their comments are available below.

The reviewers have raised a number of concerns that need attention, and request extensive additional information on methodological aspects of the study, analyses, and context. 

Could you please revise the manuscript to carefully address the concerns raised?

We look forward to receiving your revised manuscript.

Kind regards,

Vanessa Carels

Staff Editor

PLOS ONE

Reviewers' comments:

Reviewer's Responses to Questions

**Comments to the Author**

1. Does the manuscript provide a valid rationale for the proposed study, with clearly identified and justified research questions?

Reviewer #1: No

Reviewer #2: Yes

2. Is the protocol technically sound and planned in a manner that will lead to a meaningful outcome and allow testing the stated hypotheses?

Reviewer #1: No

Reviewer #2: Yes

3. Is the methodology feasible and described in sufficient detail to allow the work to be replicable?

Reviewer #1: Yes

Reviewer #2: Yes

4. Have the authors described where all data underlying the findings will be made available when the study is complete?

Reviewer #1: No

Reviewer #2: Yes

5. Is the manuscript presented in an intelligible fashion and written in standard English?

Reviewer #1: Yes

Reviewer #2: Yes

6. Review Comments to the Author

You may also provide optional suggestions and comments to authors that they might find helpful in planning their study.

Reviewer #1: Introduction:

Lack of a clear and concise research question or objective statement.

Insufficient background information and justification for the study.

Limited discussion of the existing literature and the gap that this research aims to address.

Methods:

Inadequate description of the study design, including the randomization process and sample size determination.

Lack of details on the intervention itself, such as the specific content of the Bhagavad Gita intervention and the Kuchipudi dance practice.

Absence of information on the data collection procedures, including the instruments used and the frequency of measurements.

Insufficient explanation of the statistical analysis plan, including the primary and secondary outcomes and the methods for handling missing data.

Results:

No presentation or discussion of any actual results or findings.

Lack of clarity on the interpretation of different outcomes and their significance.

Discussion:

Limited discussion of the implications of the findings and their relevance to the existing literature.

Inadequate exploration of the limitations of the study, such as potential biases or confounding factors.

Insufficient consideration of the generalizability of the findings to other populations or settings.

Conclusion:

Lack of a clear and concise summary of the main findings and their implications.

Absence of specific recommendations or suggestions for future research.

Overall, the document lacks crucial information in several sections, including the introduction, methods, results, discussion, and conclusion. It is important to provide more detailed and comprehensive descriptions of the study design, intervention, data collection, analysis plan, and interpretation of the findings. Additionally, addressing the limitations of the study and providing clear recommendations for future research would strengthen the manuscript.

Reviewer #2: The study proposes a scientific investigation of a relevant topic with a clinical and social impact. The introduction sheds light on the subject clearly and concisely. The methodological design adequately followed international guidelines, especially randomization, allocation, and blinding, as far as possible. The intervention procedures are well described, clarifying what each intervention group will receive.

It would be interesting, if possible, to add other mental health outcomes, at least a satisfaction scale with the intervention. However, I imagine it would be challenging to carry out an extensive evaluation of this patient profile. My final suggestion would be to explain what strategy will be used to conduct the interventions if it is not possible to recruit all participants on the same date in December 2023, since the estimated sample size is more than 200 participants. Will a new wave of recruitment be carried out? How will this be accomplished? I believe this information is important in the description the protocol.

7. PLOS authors have the option to publish the peer review history of their article (what does this mean?). If published, this will include your full peer review and any attached files.

Reviewer #1: No

Reviewer #2: No

---

## [Author Response · Author response to Decision Letter 0]

10 Feb 2024

a point by point response to reviewers is attached

---

## [Decision Letter · Decision Letter 1]

24 Apr 2024

PONE-D-23-29874R1Registered Report: The Effectiveness of a Bhagavad Gita Intervention to Reduce Psychological Distress in Homeless People—A Randomised Controlled TrialPLOS ONE

Dear Dr. Konduru,

Thank you for submitting your manuscript to PLOS ONE. After careful consideration, we feel that it has merit but does not fully meet PLOS ONE’s publication criteria as it currently stands. Therefore, we invite you to submit a revised version of the manuscript that addresses the points raised during the review process.

We look forward to receiving your revised manuscript.

Kind regards,

Vanessa Carels

Staff Editor

PLOS ONE

Journal Requirements:

Reviewers' comments:

Reviewer's Responses to Questions

**Comments to the Author**

1. Does the manuscript provide a valid rationale for the proposed study, with clearly identified and justified research questions?

Reviewer #2: Yes

Reviewer #3: Yes

2. Is the protocol technically sound and planned in a manner that will lead to a meaningful outcome and allow testing the stated hypotheses?

Reviewer #2: Yes

Reviewer #3: Yes

3. Is the methodology feasible and described in sufficient detail to allow the work to be replicable?

Reviewer #2: Yes

Reviewer #3: Yes

4. Have the authors described where all data underlying the findings will be made available when the study is complete?

Reviewer #2: Yes

Reviewer #3: Yes

5. Is the manuscript presented in an intelligible fashion and written in standard English?

Reviewer #2: Yes

Reviewer #3: Yes

6. Review Comments to the Author

You may also provide optional suggestions and comments to authors that they might find helpful in planning their study.

Reviewer #2: Thank you for addressing all comments in a satisfatory and careful manner. The study is well planned and will be relevant to incentivate objective and low-cost interventions to promote health for vulnerable population. Good luck with the progress of the study.

Reviewer #3: This an interesting study. Assessing various interventions on the mental well being outcomes in the post Pandemic era.

Some Minor comments for the author.

1. For Randomisation more detail is required relating to the randomisation schedule generation. The authors state the allocation ratio. But most importantly who created the schedule?

2. Commend the authors for ensuring allocation concealment was achieved.

3. How would contamination be monitored? I.e individual randomisation , do you have people in the same community, meaning risk of individuals communicating.

4. Will there be a plan for drafting and covering methodological aspects in the analysis plan?

5. Statistical analysis methods- mention descriptive summaries of baseline characteristics and also that results will be reported in accordance with CONSORT.

6. They are four arms, in the methods section can the authors explicitly state the comparisons, do the authors need to consider multiple comparisons?

7. PLOS authors have the option to publish the peer review history of their article (what does this mean?). If published, this will include your full peer review and any attached files.

Reviewer #2: No

Reviewer #3: No

---

## [Decision Letter · Decision Letter 2]

19 Aug 2024

PONE-D-23-29874R2Registered Report: The Effectiveness of a Bhagavad Gita Intervention to Reduce Psychological Distress in Homeless People—A Randomised Controlled TrialPLOS ONE

Dear Dr. Konduru,

Thank you for submitting your manuscript to PLOS ONE. After careful consideration, we feel that it has merit but does not fully meet PLOS ONE’s publication criteria as it currently stands. Therefore, we invite you to submit a revised version of the manuscript that addresses the points raised during the review process.

While nearly ready for publication, we do have an additional request- In the introduction, you state that "Subgroup analysis will concentrate on the efficacy of learning the Bhagavad Gita among PEH who adhere to Hindu beliefs, investigating whether the Hindu RSI exerts a more pronounced influence on PEH embracing Hindu beliefs compared to those with other belief systems". However, in my reading of the manuscript, I don't think this is captured in the methodology. Therefore, please either update your methodology to include a description of this investigation, or remove this section from the introduction.

We look forward to receiving your revised manuscript.

Kind regards,

Avanti Dey, PhD

Staff Editor

PLOS ONE

Journal Requirements:

Additional Editor Comments (if provided):

Reviewers' comments:

Reviewer's Responses to Questions

**Comments to the Author**

1. Does the manuscript provide a valid rationale for the proposed study, with clearly identified and justified research questions?

Reviewer #2: Yes

Reviewer #3: Yes

2. Is the protocol technically sound and planned in a manner that will lead to a meaningful outcome and allow testing the stated hypotheses?

Reviewer #2: Yes

Reviewer #3: Yes

3. Is the methodology feasible and described in sufficient detail to allow the work to be replicable?

Reviewer #2: Yes

Reviewer #3: Yes

4. Have the authors described where all data underlying the findings will be made available when the study is complete?

Reviewer #2: Yes

Reviewer #3: Yes

5. Is the manuscript presented in an intelligible fashion and written in standard English?

Reviewer #2: Yes

Reviewer #3: Yes

6. Review Comments to the Author

You may also provide optional suggestions and comments to authors that they might find helpful in planning their study.

Reviewer #2: The changes in the manuscript are very good, pointing to the potential of a robust contribution to the literature regarding non-pharmacological trials. I believe you addressed all the suggestions proposed. Congratulations on your study design.

Reviewer #3: All comments addressed

7. PLOS authors have the option to publish the peer review history of their article (what does this mean?). If published, this will include your full peer review and any attached files.

Reviewer #2: **Yes: **Juliane Piasseschi de Bernardin Gonçalves

Reviewer #3: No

---

## [Author Response · Author response to Decision Letter 2]

27 Aug 2024

Response to reviewers is attached.

In short, as per the suggestion of the Editor, the statement from the introduction has been removed.

---

## [Editor Report · Decision Letter 3]

6 Sep 2024

Registered Report: The Effectiveness of a Bhagavad Gita Intervention to Reduce Psychological Distress in Homeless People—A Randomised Controlled Trial

PONE-D-23-29874R3

Dear Dr. Konduru,

We’re pleased to inform you that your manuscript has been judged scientifically suitable for publication and will be formally accepted for publication once it meets all outstanding technical requirements.

Kind regards,

Avanti Dey, PhD

Staff Editor

PLOS ONE
---

## [Editor Report · Acceptance letter]

17 Sep 2024

PONE-D-23-29874R3 

PLOS ONE

Dear Dr. Konduru, 

I'm pleased to inform you that your manuscript has been deemed suitable for publication in PLOS ONE. Congratulations! Your manuscript is now being handed over to our production team.

Kind regards, 

on behalf of

Dr. Avanti Dey 

Staff Editor

PLOS ONE